# The Impact of Ambient Atmospheric Mineral-Dust Particles on the Calcification of Lungs

Mariola Jabłońska [ID], Janusz Janeczek *[ID] and Beata Smieja-Król [ID]

University Laboratories of Atmospheric Survey, Faculty of Natural Sciences, Institute of Earth Sciences, University of Silesia, 60 Będzińska St., 41-200 Sosnowiec, Poland; mariola.jablonska@us.edu.pl (M.J.); beata.smieja-krol@us.edu.pl (B.S.-K.)

* Correspondence: janusz.janeczek@us.edu.pl; Tel.: +48-323689261

**Abstract:** For the first time, it is shown that inhaled ambient air-dust particles settled in the human lower respiratory tract induce lung calcification. Chemical and mineral compositions of pulmonary calcium precipitates in the lung right lower-lobe (RLL) tissues of 12 individuals who lived in the Upper Silesia conurbation in Poland and who had died from causes not related to a lung disorder were determined by transmission and scanning electron microscopy. Whereas calcium salts in lungs are usually reported as phosphates, calcium salts precipitated in the studied RLL tissue were almost exclusively carbonates, specifically Mg-calcite and calcite. These constituted 37% of the 1652 mineral particles examined. Mg-calcite predominated in the submicrometer size range, with a $MgCO_3$ content up to 50 mol %. Magnesium plays a significant role in lung mineralization, a fact so far overlooked. The calcium phosphate (hydroxyapatite) content in the studied RLL tissue was negligible. The predominance of carbonates is explained by the increased $CO_2$ fugacity in the RLL. Carbonates enveloped inhaled mineral-dust particles, including uranium-bearing oxides, quartz, aluminosilicates, and metal sulfides. Three possible pathways for the carbonates precipitation on the dust particles are postulated: (1) precipitation of amorphous calcium carbonate (ACC), followed by its transformation to calcite; (2) precipitation of Mg-ACC, followed by its transformation to Mg-calcite; (3) precipitation of Mg-free ACC, causing a localized relative enrichment in Mg ions and subsequent heterogeneous nucleation and crystal growth of Mg-calcite. The actual number of inhaled dust particles may be significantly greater than was observed because of the masking effect of the carbonate coatings. There is no simple correlation between smoking habit and lung calcification.

**Keywords:** air pollution; lung mineralization; magnesian calcite; calcite; amorphous calcium carbonate

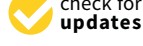

## 1. Introduction

Pulmonary calcification involves the precipitation of calcium salts in lung tissues. The numerous causes of pathogenic calcification are grouped into metastatic, dystrophic, and idiopathic causes [1–3]. Metastatic calcification, both benign and malignant, refers to calcium deposition in normal tissues caused by high levels of serum calcium and phosphate, whereas dystrophic calcification refers to the deposition of calcium salts in previously injured cells and tissues. Dystrophic calcification refers to calcification triggered by asbestosis, silicosis, coal-miner's pneumoconiosis, and other occupational diseases caused by inhaled mineral dust [1,2]. Metastatic calcification is composed chiefly of either amorphous or microcrystalline whitlockite, $(Ca,Mg)_3PO_4$, with subordinate pyrophosphate [4]. According to Farver [3], calcium is usually present in the form of calcium phosphate. Dystrophic calcification is a localized process that leads to the deposition of crystalline hydroxyapatite (HAP) [1]. The rare idiopathic lung disorder known as pulmonary alveolar microlithiasis causes the deposition of micronodules composed of either calcium phosphate [5,6] or calcium carbonates [2].

Lung calcification is routinely diagnosed by X-ray chest radiography or computed tomography. These techniques do not enable the mineral species that are building the

calcium "lesions" to be determined. Hence, usually no attempts are made to distinguish between the various possible calcium salts deposited. Instead, morphological aspects of calcification are important in etiology and treatment.

While the adverse effect of large numbers of mineral particles on human cardiopulmonary health is well known [7–15], there have been no attempts to relate lung calcification to inhaled ambient atmospheric dust particles. Moreover, Ca-bearing particles encountered in lung tissues are assumed to be indicators for tobacco smoke [16]. In this paper, by examining the mineral composition of lung right lower-lobe (RLL) tissues of 12 individuals from the Upper Silesia conurbation (USC), Poland, we show that endogenous particles of Ca- and (Ca, Mg)-carbonates in lung tissues may originate in response to the settlement of inhaled atmospheric dust particles.

## 2. Mineral Inventory of Inhaled Dust Particles

Mineral assemblages found in lungs are either endogenous, i.e., formed biogenically in situ, or exogenous (inhaled). The former includes Ca-carbonates, Ca-phosphates, and, rarely, Ca-oxalates [2]. Quantities of mineral particles inhaled by people not occupationally exposed to dust can be quite high. Approximately $10^{11}$ ultrafine (<0.1 µm in diameter) dust particles may be deposited daily in the respiratory tract of a person living in the Los Angeles, California area [17]. The life-time accumulation of mineral dust particles in the lungs of a non-occupationally exposed North American urban dweller is on average $0.5 \times 10^9$ particles per gram of dry lung [18]. The geometric mean of total particle contents in the lungs of Mexico City, Mexico inhabitants is $2.055 \times 10^6$ particles/g of dry lung, but is approximately ten times less in those of residents in Vancouver, Canada [16].

Brauer et al. [16] examined the parenchymal particle contents of autopsy lungs from never-smoking female residents of Mexico City and Vancouver. From their observations, 96% of the retained particles were <2.5 µm in aerodynamic diameter, including aggregated ultrafine particles. Brauer et al. [16] concluded that individuals residing for a long time (>60 years) in an area of high ambient particle concentrations retain much greater numbers of those particles compared to long-term residents of an area of low ambient pollution levels.

The mineral composition of dust inhaled due to occupational exposures is well established [19,20]. However, less is known about the mineral composition of dust particles deposited in the lungs of those who were not occupationally exposed to dust. The pioneering studies in this respect were by Churg and Wiggs [21] and Churg et al. [22] in Vancouver, Paoletti et al. [23] in the Rome area, and Stettler et al. [18] in Cincinnati, Ohio. The majority of particles observed in these studies were rock-forming minerals, including free silica (predominantly quartz), aluminosilicates (feldspars), and layered silicates (clays, kaolinite, micas, talc).

Microanalysis of lung autopsy tissues of individuals exposed to the catastrophic London smog episode in December 1952 revealed aggregates of ultrafine (≤0.1 µm) carbonaceous material associated with fine (≤1.0 µm) heavy-metal-bearing particles [24]. Domingo-Neumann et al. [25] analyzed the ashed lung samples of 30 subjects from Fresno, California, and found that the mineral composition of dust particles (quartz, plagioclase, K-feldspars, biotite) was similar to that of particulate matter with an aerodynamic diameter < 10 µm ($PM_{10}$) sampled during field agricultural operations in that region. A recent SEM study of 33 lung-tissue samples from individuals exposed to desert dust and other sources of inhaled particulate matter revealed 51 distinct phases among 13,000 particles examined; the most common phases were of silica (a collective term for both crystalline and amorphous silicon dioxide), feldspars, apatite, titanium oxides, and iron oxides [26].

As expected, all of the sources quoted above show that the mineral inventory of exogenous particles settled in human lungs reflects that of dust particles in ambient air. Not surprisingly, the most abundant particles are of natural aluminum silicates (feldspars and mica) and silica. These are typical constituents of continental crust and, as such, they are the predominating dust particles in non-industrial regions. The relative abundance of

titanium oxides (Cincinnati) and talc (Cincinnati, Rome, Vancouver) resulted from their wide use in a range of industrial and consumer products. All of the observed minerals were emitted either from natural sources or were released due to anthropogenic activities.

## 3. Site Characterization

The Upper Silesian conurbation (USC) is located in the center of the Silesian province (voivodship) in southern Poland. The USC consists of 14 adjacent cities with a total population of 1.85 million [27]. Together with an additional 15 directly bordering communities, the USC forms one of the most urbanized and industrialized regions of Europe, with ca. 2.5 million people. The USC is the most densely populated region of Poland (on average, 887 persons/km$^2$ and 3785 persons/km$^2$ in one of the cities). Despite significant improvement in air quality since the beginning of political and economic transformations in Poland in 1989, air pollution in the USC still causes serious environmental and health problems. Residential coal-burning, the major emission source of air pollutants in the region, accounted for 66 wt.% of PM$_{10}$, 76 wt.% of PM$_{2.5}$, and 94 wt.% of benzo[a]pyrene in 2018 [28]. Industrial sources (electricity- and heat-generating coal-fired plants, metal manufacturers including steel mills, zinc and copper smelters and refineries, coking and chemical plants, coal mines) account for 14 wt.% and 15 wt.% of PM$_{10}$ and PM$_{2.5}$, respectively [28]. Unlike in other European metropolitan areas, the vehicular transport contribution to air pollution is relatively low (5 wt.% PM$_{10}$ and 4 wt.% PM$_{2.5}$) despite the large number of motor vehicles and extensive network of high-traffic roads [28].

Between 2010–2018, the PM$_{2.5}$ mean annual concentrations in all cities of the USC exceeded the annual limit value of 25 μg/m$^3$ [27]. The seasonal peak in airborne particle concentrations occurs during the heating season (usually late October to late April); it is caused by coal- and biomass-burning for domestic heating. The PM$_{2.5}$ concentration in winter is twice that in summer. For instance, the average PM$_{2.5}$ concentration during the winter of 2018 in Katowice, the capital of the USC, was 47 μg/m$^3$, compared to 22 μg/m$^3$ in summer [27]. In winter, the air quality is classified as unhealthy for sensitive groups, and unhealthy or very unhealthy for more than 60% of the time. During unfavorable weather conditions (stagnant dry air, temperature inversion), the air quality may reach the hazardous category. The seasonality of peak ambient air pollution is reflected in the seasonal exacerbation of respiratory diseases [29]. The daily average mortality due to cardio-respiratory disorders in the USC is significantly higher in winter than in summer, and it correlates well with high daily PM$_{2.5}$ concentrations [30]. A relationship between an increased morbidity to lung cancer in males and elevated concentrations of PM$_{10}$ in ambient air has been observed in Upper Silesia [31]. Some 20% of those with diagnosed lung cancer in Upper Silesia were non-smokers (never smoked) and had never been exposed occupationally to carcinogenic substances. Thus, they may have been environmentally exposed to respirable dust and other carcinogenic agents.

Most of the constituents of ambient atmospheric dust in the USC are anthropogenic, including rock-forming minerals that may have been released during mineral processing or construction [32]. Major airborne dust phases include: quartz (>20 vol. %), soot (<10 vol. % in summer and up to 90 vol. % in winter), fly-ash (20 vol. % throughout the year), gypsum (20 vol. % in winter), and Fe-oxides (hematite, magnetite, wüstite). In addition to the major constituents, over 30 minor- or accessory mineral phases have been identified in atmospheric dust samples in the USC [32–34].

A comparison of the exogenous-particle inventory observed in samples of the lung right lower-lobe (RLL) of 12 individuals who lived in the USC (Table 1) with that of atmospheric particulate matter reveals a close similarity with the notable exceptions of soot and graphite [33]. Though soot and graphite commonly occur in the USC's atmospheric particulate matter, they have not been observed in RLL [33]. Chemical compositions and particle morphologies encountered in atmospheric dust and in lungs are identical. However, the assemblages differ in relative mineral proportions. Calcite and its Mg-rich variety are predominant in lung samples, whereas carbonates, with dolomite dominating,

are subordinate components (<8 vol. %) of the airborne particulate matter, and they have distinctly different morphologies [34].

**Table 1.** Mineral composition, number (*n*), content (vol. %), and sizes of particles observed in RLL tissues of 12 individuals in the USC. Compiled and modified from data in [33].

| Mineral Phase | *n* | Content (vol. %) | | Size (μm) | |
|---|---|---|---|---|---|
| | | Mean | Range | Mean * | Range |
| Carbonates | 612 | 32 | 5.5–65 | 1.42(06) | 0.22–5.05 |
| **Al-silicates** [1] | 243 | 19 | 12–34 | 2.10(12) | 0.26–5.01 |
| **Silica** [2] | 149 | 10 | 3–17 | 1.90(12) | 0.35–4.65 |
| **Fe-oxides** [3] | 199 | 12 | 5–30 | 1.67(11) | 0.31–4.98 |
| **Halides** [4] | 112 | 7 | 2–9 | 1.69(15) | 0.54–5.04 |
| **Iron** | 35 | 2 | 1–4 | 1.74(36) | 0.42–4.93 |
| **Sulfides** [5] | 53 | 3 | 1–5 | 1.36(22) | 0.34–4.08 |
| **Metals** [6] | 49 | 3 | 0.4–7 | 1.72(29) | 0.22–5.02 |
| Spinels [7] | 38 | 3 | 1–4 | 1.83(31) | 0.17–5.08 |
| Alloys [8] | 39 | 2 | 1–4 | 1.58(30) | 0.25–4.33 |
| Oxides [9] | 53 | 2 | 0–7 | 1.13(17) | 0.17–4.46 |
| Gypsum | 26 | 2 | 2–5 | 2.11(32) | 0.68–4.76 |
| **Barite** | 32 | 2 | 1–5 | 1.25(22) | 0.42–3.60 |
| Ca-phosphate | 9 | 1 | 1–3 | 2.70(70) | 0.68–4.76 |
| REE-phosphates | 3 | 0.5 | 0.3–0.9 | 1.47(26) | 0.97–2.08 |

Mineral phases observed in lung tissues of all 12 subjects are in bold; * numbers in brackets refer to the standard deviation; [1] include spherical amorphous (vitreous) aluminosilicates (fly-ash) (ca. 20 vol. %), feldspars, mullite, amphiboles, clay minerals (illite, talc); [2] quartz, tridymite, and amorphous silica (ca.11 %); [3] hematite, magnetite, wüstite, goethite, ferrihydrite; [4] halite, sylvite; [5] galena, sphalerite, and Fe-sulfides; [6] Au, Sn, Pb, Ti; [7] magnesio-ferrite, franklinite, hercynite, jacobsite; [8] Cu-Ni-Zn alloys and steel; [9] oxides other than spinels and Fe-oxides: brookite ($TiO_2$), Sn, Zr, Zn-oxides.

## 4. Materials and Methods

Autopsy samples of lung tissues from 12 USC residents (four females and eight males), aged 18 to 89 years old at time of death not related to lung diseases, were provided by the DiagnoMed company and a clinical hospital in Zabrze together with information on age, gender, smoking habits and medical records of each subject. Ten individuals were never occupationally exposed to dust. Two individuals were retired coal miners aged 81 and 87. Seven individuals, including the former coal miners, were life-long smokers. None of the individuals had a record of lung disorders. All samples were from the RLL and consisted of non-tumoral subpleural tissues. The size of each sample was 2 cm ×2 cm × 0.5 cm.

The samples, after drying at 37 °C, were examined using transmission electron microscopy (TEM) and scanning electron microscopy (SEM). Samples for TEM were prepared by gently crushing fragments of lung tissues in a mortar and placing them on a standard copper mesh grid. Observations and analyses were performed using JEOL JEM-2000FX (accelerating voltage 200 keV, JEOL Ltd., Tokyo, Japan) and JEOL JEM-3010 (accelerating voltage 300 keV, JEOL Ltd., Tokyo, Japan) microscopes. Phase identification was done by combining energy-dispersive X-ray spectrometry (EDX) and selected area electron diffraction (SAED) data. Secondary electron (SE) and back-scattered electron (BSE) images of uncoated samples were acquired using a Phillips 30XL (FEI, Brno, Czech Republic) environmental analytical scanning electron microscope equipped with an EDAX EDS Sapphire system operated at 15 keV accelerating voltage. Pressure in the sample chamber was 0.3 Torr. Some samples were carbon-coated and observed using an Inspect F high-vacuum scanning electron microscope with field emission operated at accelerating voltages of 15 and 20 keV.

The size of individual mineral particles was measured in two dimensions using calibrated SEM and TEM images and Visio Drawing 2000 (Microsoft) software. The equivalent diameter of particles was determined as a root-mean-square size, $\{a^2 + b^2\}^{1/2}$, where *a* and *b* are the maximum and minimum orthogonal distances between points on

the particle perimeter. Based on those measurements, the volume concentration ($V_c$) was calculated from the formula:

$$Vc = \frac{m}{n} \times 100\% \pm u_\alpha \sqrt{\frac{\frac{m}{n}\left(1 - \frac{m}{n}\right)}{n}} \times 100\% \qquad (1)$$

where $m$ is the sum of equivalent diameters of all particles of a given mineral species, $n$ is the sum of equivalent diameters of all mineral particles in the sample, and $u_\alpha$ is the critical value of the distribution function for a given confidence level.

Mineral-saturation indices were computed using the PHREEQC software (version 3, U.S. Geological Survey, Reston, Virginia, USA) [35] for both human blood plasma and simulated lung fluid (Gamble's solution).

## 5. Results and Discussion

Carbonates constituted 37% of the 1652 mineral particles examined in the investigated samples. Carbonate contents varied from 5.48 vol. % in the tissues of an 18-year-old smoking female to 64.62 vol. % in the tissues of an 82-year-old non-smoking female. The mean value was 31.94 ± 0.29 vol. % (Table 2). The highest concentration of carbonates among males (43.59 vol. %) was observed in the tissues of an 87-year-old life-long smoker. The highest number of carbonate particles (102) associated with the highest number of mineral particles (289) was observed in the tissues of a 54-year-old male smoker. The quantities of carbonates in RLL of non-smoking females was higher than that those in both smoking- and non-smoking males of the same or similar age (Table 2). Among males, the amounts of carbonates tended to be higher in the tissues of non-smoking individuals. A plausible explanation for this observation is that smoking causes a decrease in the pH of lung fluids [36], inhibiting precipitation of carbonates. This accords with the measured elevated bulk concentrations of Ca in the RLL of non-smoking individuals (3000–8000 ppm) compared to smokers (<1500 ppm; [33]).

Carbonates occurred as clusters of polycrystalline aggregates ranging from 0.5–7 μm in diameter or as single crystals ranging from 0.2–5 μm in length (Figure 1). The largest (7 μm) particles were agglomerates of smaller subgrains. The mean equivalent diameter of the carbonate particles was 1.42 ± 0.06 μm ($n$ = 612; variation coefficient = 57%). Single particles had smooth surfaces and sharp edges (Figure 1).

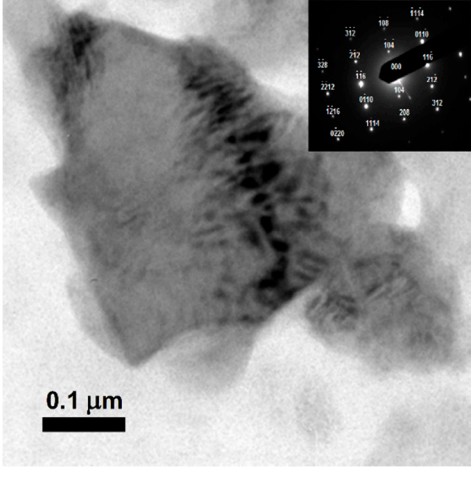

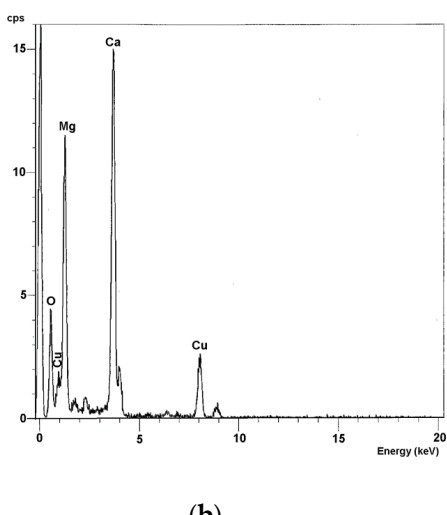

(**a**)  (**b**)

**Figure 1.** (**a**) TEM image, SAED pattern, and (**b**) EDS spectrum of Mg-calcite crystal from RLL tissue of a 30-year-old male smoker.

**Table 2.** Contents of carbonate particles (CP) related to the total number of mineral particles (TP) and sizes (µm) of CP in RLL tissues of 12 individuals from the USC.

| Age | Females | | | | Males | | | | | | | |
|---|---|---|---|---|---|---|---|---|---|---|---|---|
| | 18s | 54ns | 65ns | 82ns | 30s | 46s | 53ns | 54s | 74s | 81sm | 82ns | 87sm |
| CP/TP | 6/83 | 49/117 | 66/112 | 77/108 | 49/211 | 29/158 | 24/113 | 102/289 | 28/85 | 95/201 | 26/68 | 46/107 |
| vol. % | 5.48 | 36.05 | 41.57 | 64.62 | 21.06 | 18.65 | 21.09 | 28.50 | 29.20 | 38.29 | 40.09 | 43.59 |
| Diameter Range | 0.68–2.48 | 0.37–4.46 | 0.23–3.51 | 0.38–5.05 | 0.30–4.45 | 0.22–4.98 | 0.44–3.14 | 0.23–4.43 | 0.39–4.25 | 0.22–3.58 | 0.52–3.97 | 0.56–3.90 |

Note: s—smoker; n—nonsmoker; m—coal miner.

Unlike the ubiquitous carbonates, only nine calcium phosphate (hydroxyapatite, HAP) particles were found in the tissues of five male individuals, four of whom were life-long smokers. The maximum number of the observed HAP particles in a single sample was three in tissues of two individuals who were 30 and 54 years old. Samples of three other individuals contained single particles. The equivalent diameter of the HAP particles ranged from 2.02 to 3.77 μm. Volumetrically, HAP constituted only 1.25–2.76 % of all mineral particles, and HAP/carbonate values ranged from 0.02–0.06. The lesser occurrence of HAP relative to that of carbonates was further reflected in the lack of correlation between the bulk Ca and P concentrations in RLL [33].

Four carbonate species, namely, amorphous calcium carbonate (ACC), calcite ($CaCO_3$), Mg-calcite ($Ca_{1-x}Mg_xCO_3$), and dolomite ($CaMg(CO_3)_2$) were identified by SAED and EDS. Electron-diffraction patterns of the Mg-calcite particles matched the standard (am-csd_0001327) of biogenic magnesium calcite. The hexagonal unit cell parameters calculated from $d_{3\overline{2}4}$ (1.515 Å) and $d_{\overline{1}38}$ (1.283 Å) were: $a$ = 4.966 Å, $c$ = 16.714 Å, $V$ = 412.19 Å$^3$. These were smaller than the unit cell parameters calculated for the amcsd_0001327 standard ($a$ = 4.972 Å, $c$ = 16.927 Å, $V$ = 418.45 Å$^3$).

The number of dolomite particles in the RLL tissues ranged from 2–6 per sample, i.e., <10% of all carbonate particles. They were distinguished from Mg-calcite by SAED patterns. Airborne dolomite, a common constituent of dust particles in the USC, originates from the quarrying of dolomite deposits and from the natural weathering of dolomite outcrops in the region [34]. Thus, the dolomite particles were considered to be exogenous in the lung tissues.

Out of 612 observed carbonate particles, 64 (10.5%) were submicrometer (<1 μm) in size. Mg-calcite was the most abundant (50 particles, 78%) submicrometer carbonate (Figure 1), with Mg-free calcite constituting the remainder (14 particles, 22%). Together, calcite and Mg-calcite constituted 26% of all submicrometer particles in the RLL tissues examined (Figure 2).

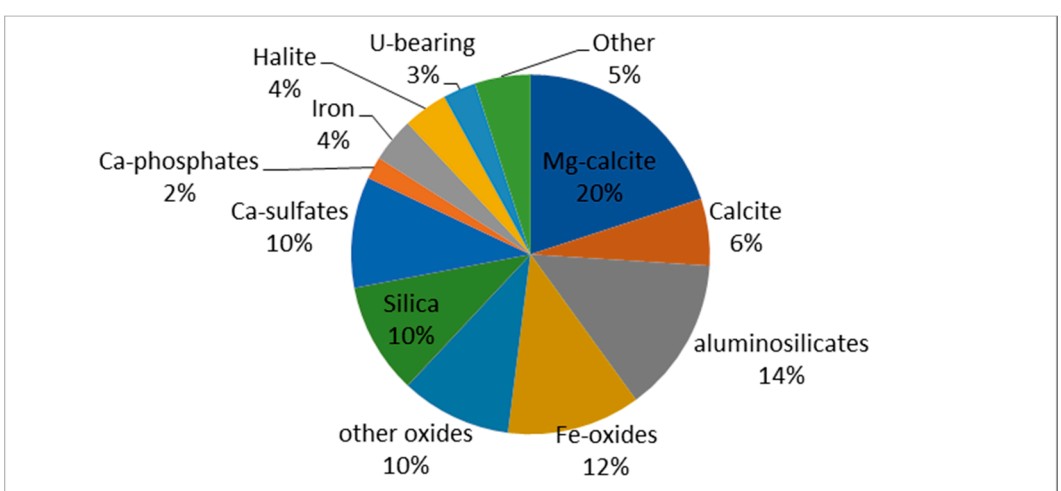

**Figure 2.** Mineral composition of submicrometer particles in RLL tissues of 12 individuals from the USC.

The values of the $K_\alpha Mg/K_\alpha Ca$ peak intensity ranged from 0.33–1.13 (mean 0.75) in the EDX spectra of 48 Mg-calcite particles, which suggests a wide range of Mg contents in the Mg-calcite. Due to technical problems (rough particle surfaces, small size), obtaining reliable quantitative data on the chemical composition of the Mg-calcite proved difficult. Standardless semi-quantitative EDX microanalysis of Ca and Mg in 5 carbonate particles gave $MgCO_3$ mole percentages of 29, 31, 47, 52, and 53. These values are typical for very high magnesian calcite (30–45 mol % $MgCO_3$) and disordered dolomite, i.e., proto-dolomite with 46–50 mol % $MgCO_3$ [37]. From the molar Mg/Ca value in a simulated lung fluid (SLF) composition provided by Taunton et al. [38], which is identical to the Mg/Ca in

human blood plasma 0.60 [39], the MgCO$_3$ content in Mg-calcite precipitated from either SLF or blood plasma is expected to be 37.5 mol %.

There was a positive correlation between the amounts of carbonates related to the total of mineral particles and the age of individuals (Figure 3). This correlation was particularly strong in samples from females, despite a limited dataset. The steeper slope of the correlation line suggests that at ages > 40 years, the carbonate accumulation rate is higher in females than in males. In the RLL of the elderly, carbonates occurred as clusters of numerous particles covering large portions of the observed tissues (Figure 4). The increase in the quantity of carbonates with age relative to the amount of exogenous mineral particles suggests that precipitation of the carbonates may have been induced by inhaled dust particles. This suggestion was confirmed by dust particles that are embedded in carbonates.

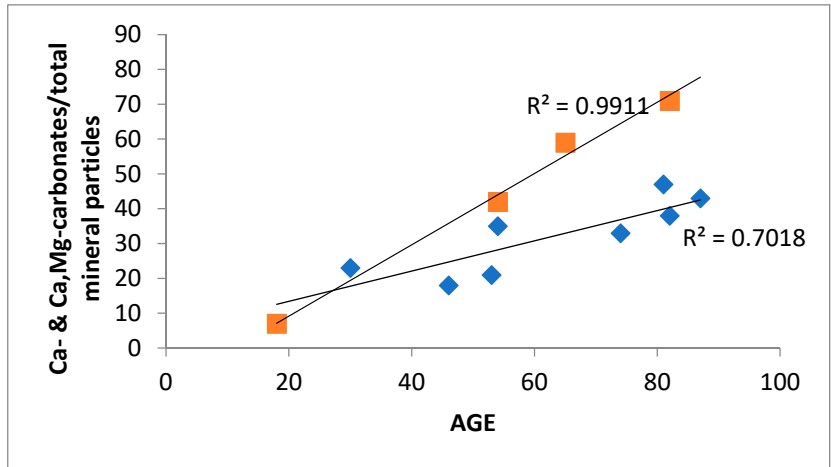

**Figure 3.** Ratio of carbonates to the total amount of mineral particles in RLL tissues of 12 individuals from the USC vs. their age (squares—females; diamonds—males).

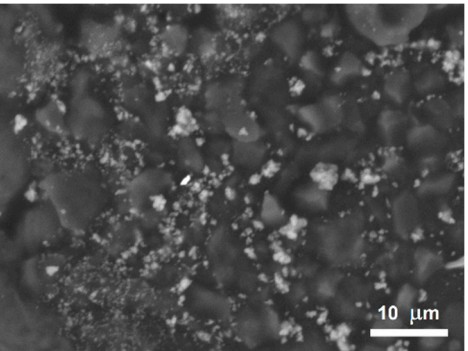

**Figure 4.** SEM image of Mg-calcite and calcite incrustations (bright grains) on RLL tissues of an 81-year-old former miner and life-long smoker.

Calcite often envelops particles of Fe-, Zn- and Pb-sulfides [33]. During this study, submicrometer particles of quartz and aluminosilicates enclosed in crystalline Mg-calcite were observed in RLL tissues (Figure 5). These observations show that the exogenous mineral particles served as nuclei for the precipitation of endogenous carbonates.

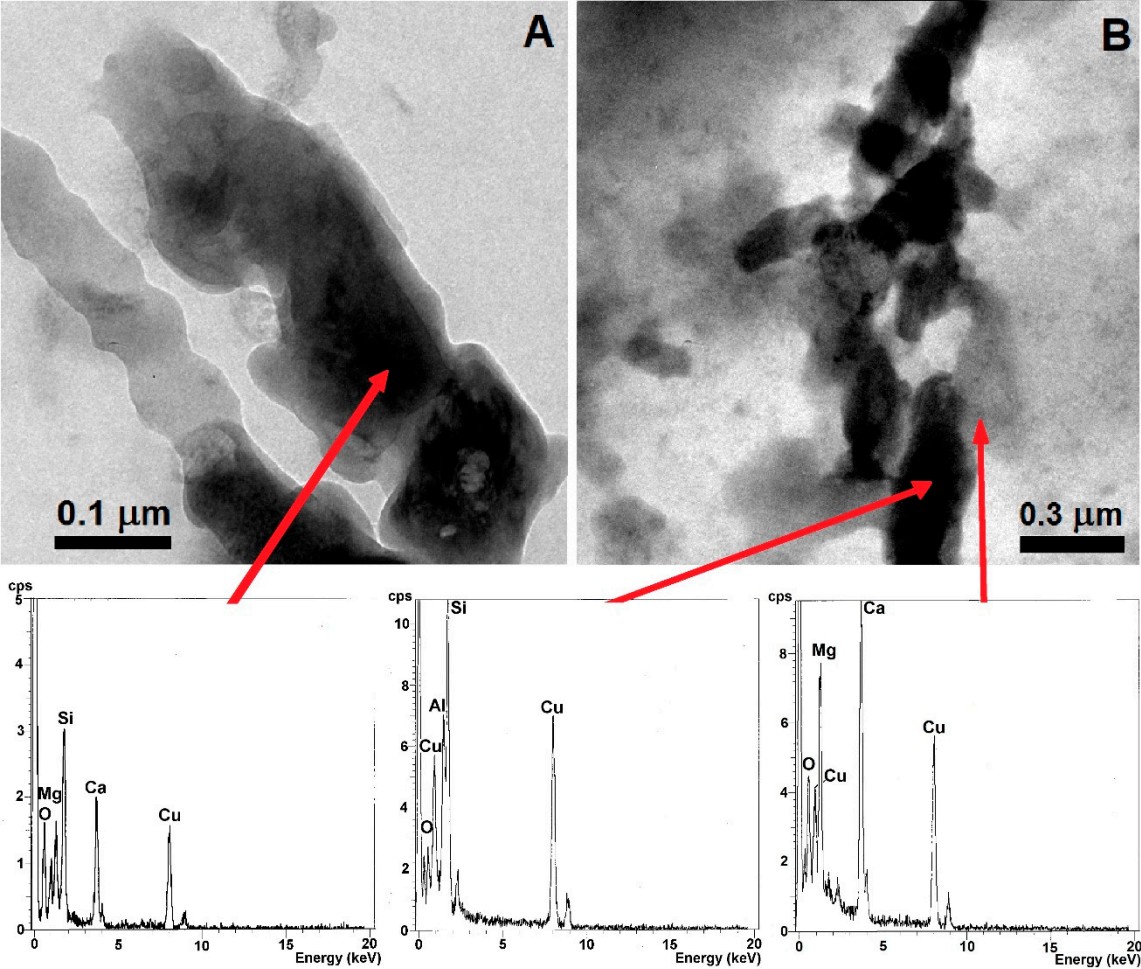

**Figure 5.** TEM images and related EDX spectra of Mg-calcite-coated: (**A**) silica and (**B**) aluminosilicate particles deposited in RLL of individuals from the USC.

Of particular interest was the observation of uranium oxide (perhaps, uraninite) and Fe-rich particles coated by ACC (Figure 6), suggested by the diffuse halos in the SAED patterns. Uranium-bearing particles, often associated with soot, have been observed in airborne mineral dust samples collected at several sites in the USC (Figure 6A). Their occurrence resembles uraninite nanocrystals encapsulated in carbonaceous matter in suspended particulates released from coal-fired power plants in Detroit, Michigan [40]. Thus, it is no surprise to find uranium-bearing particles in the lungs of people living in places affected by coal combustion. The strong Fe peak in the EDX spectrum of a U-bearing particle in lung tissue may reflect Fe-bearing phase/phases either associated with the inhaled U-bearing particle (Figure 6A) or may have originated in the process of phagocytosis [41].

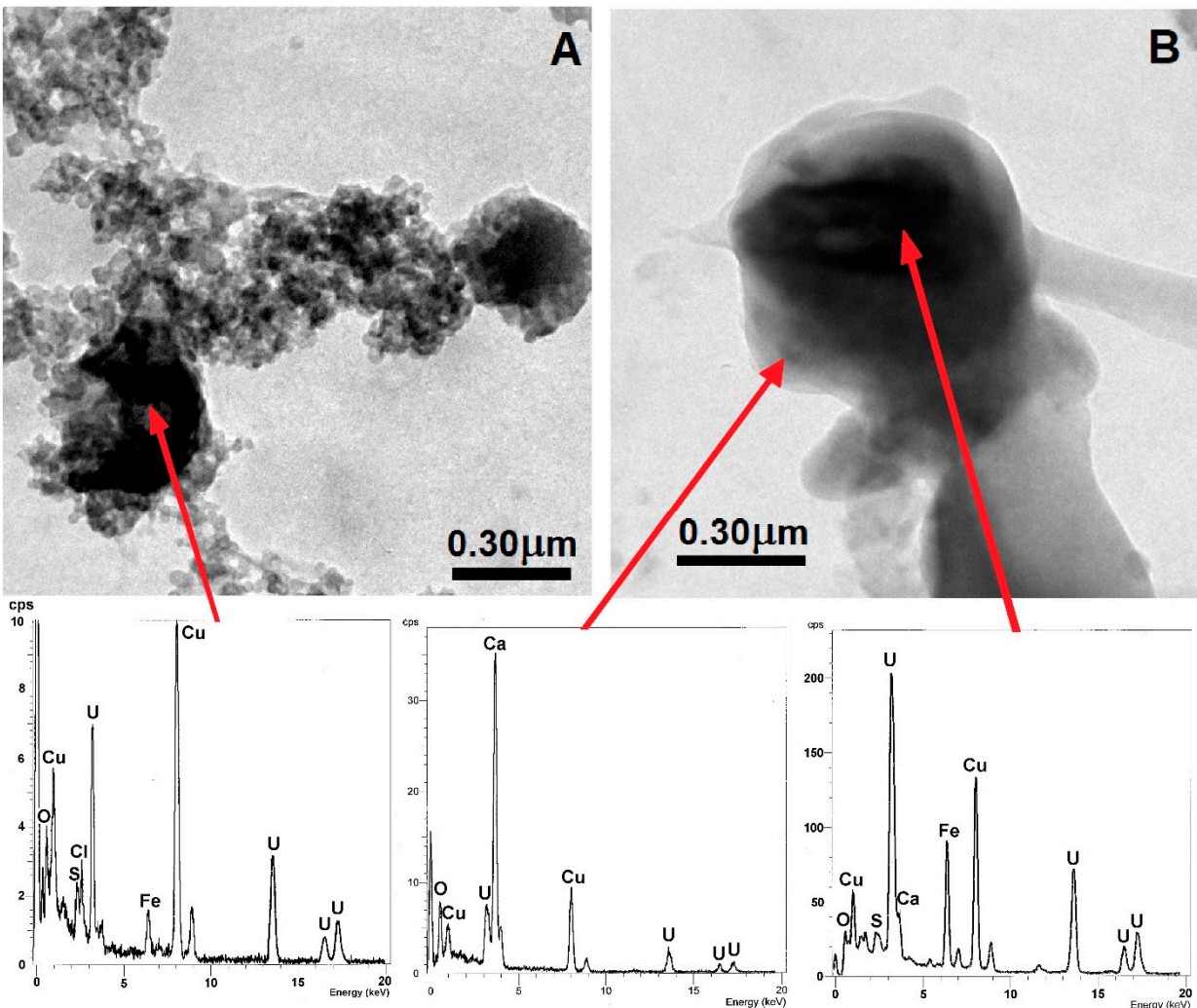

**Figure 6.** TEM images and related EDX spectra of (**A**) uranium-oxide particles and Fe-phase attached to the airborne soot aggregate collected in the USC, and (**B**) uranium-oxide particle and Fe-phase both coated by ACC in RLL tissue of a 62-year-old male non-smoker.

The ACC coating on the uranium-bearing particle (Figure 6B) may have been a precursor of calcite, according to current models for biogenic calcite formation [42–45]. The biogenic formation of calcite is a two-stage process that involves initial precipitation of thermodynamically unstable ACC, which spontaneously and rapidly crystallizes to calcite either directly or through a nanocrystalline vaterite intermediate. The transformation of vaterite to calcite is a sluggish process, ten times slower than the transition from ACC to vaterite. However, in the presence of organic matter and/or Mg, calcite is stabilized over vaterite [43]. No evidence for vaterite was observed during this study.

The predominance of Mg-calcite among submicrometer carbonates suggests that their amorphous precursor was also enriched in Mg. Laboratory experiments show a 1:1 dependence on the Mg/Ca ratio in solution and carbonate precipitated from that solution [44,46]. However, the EDX spectra of the ACC coating of U-bearing particles (Figure 6) did not show X-ray peaks from Mg.

We hypothesize that there are three possible carbonate-precipitation pathways in lungs triggered by exogenous dust particles: (1) precipitation of ACC on dust particles, followed by its transformation to calcite; (2) precipitation of Mg-ACC, followed by its transformation to Mg-calcite; and (3) precipitation of Mg-free ACC, followed by Mg-calcite heterogeneous nucleation and crystal growth. The last pathway requires a localized increase in Mg/Ca in the lung fluid due to calcium consumption by the ACC. The relative enrichment in Mg

may enable the direct precipitation of Mg-calcite on the Mg-free ACC. It is possible that localized specific biochemical conditions regulate the selection of these pathways in the process of phagocytosis.

While calcium salts precipitated in lungs are most frequently identified as phosphates (hydroxyapatite and/or whitlockite) [1–4], calcium carbonate is also reported in numerous cases. Fine acicular crystals of calcite have been found in lung tissue of individuals affected by berylliosis, sarcoidosis and tuberculosis [47]. Calcium carbonate has been found in the so-called pulmonary "blue bodies"—roughly spherical laminated structures in the cytoplasm of alveolar macrophages [48]. Under specific chemical conditions, other types of calcium salts may form in human lungs. Calcium monohydrate oxalate (mineral whewellite, $CaC_2O_2 \cdot H_2O$) may occur in lung tissue infected with *Aspergillus* species [47]. Thus, the type of calcium salt precipitated in lungs depends on the local biochemical conditions.

In response to the intrusion of a foreign body into the alveolar cells of lungs, there is a substantial increase in intracellular $Ca^{2+}$ released from both mitochondria and the endoplasmic reticulum [2,49,50]. The influx of $Ca^{2+}$ is sufficient to enable precipitation of either apatite or calcite. Thus, carbonate and phosphorous ions compete for Ca in the lung fluids.

Experiments using simulated lung fluids (SLF) at pH 7.4 and 37 °C show that HAP is a preferred precipitate because the initial SLF is supersaturated with respect to HAP (saturation index SI = log(ion activity product/solubility product) = 6.9) [38]. Taunton et al. [38] confirmed their calculations with observations using SEM/EDS on large (30 μm × 10 μm) calcium phosphate particles comprising the predominant phase in calcified pleural plaques in the lungs of 85-year-old male welder and smoker. However, the concentration of $HCO_3^-$ (0.0321 mol/L) in the SLF used by Taunton et al. [38] was an order of magnitude greater than that of $HPO_4^{2-}$ (0.00104 mol/L).

The saturation index calculated for calcite from compositions of SLF (Gamble's solution) and human blood plasma [39] during our study was 1.12 and 0.70, respectively (Table 3). Interestingly, both fluids are even more supersaturated (SI = 1.43 and 2.10, respectively) with respect to dolomite, which can be considered as a compositional analogue for Mg-calcite. While saturation index values for both carbonates were lower than for HAP (SI = 7.77 in Gamble's solution and 6.52 in blood plasma), they were positive, making precipitation of carbonates theoretically possible. Therefore, based solely on the chemical compositions of human blood plasma and SLFs, both precipitation pathways, i.e., phosphatic and carbonatic, are possible. Whether calcium is bound to phosphate or to carbonate precipitates apparently depends on the $CO_2$ fugacity ($P_{CO2}$). According to Chan et al. [2], both the high ventilation (movement of air into and out of the lungs) to perfusion (flow of blood in the pulmonary capillaries) ratio of 3.3 resulting in relatively low $P_{CO2}$ (ca 30 mm Hg) and a blood pH of 7.51 predispose the apexes of lungs to the precipitation of calcium phosphate. In the lower lobes, the pH is slightly less alkaline (7.39) due to higher $P_{CO2}$ (ca 44 mm Hg), and the ventilation-to-perfusion ratio is much lower (0.63; [51]). The increase in $P_{CO2}$ in lower lobes favors precipitation of carbonates because most of the $CO_2$ reacts with water in the presence of the catalyst carbonic anhydrase in erythrocytes [52] to produce weak carbonic acid that dissociates into the bicarbonate ion:

$$CO_2 + H_2O \leftrightarrow H_2CO_3 \leftrightarrow HCO_3^- + H^+ \tag{2}$$

**Table 3.** Saturation index (SI), ion activity product (IAP), and solubility product (Ksp) for carbonates and hydroxyapatite (HAP) computed using the PHREEQC software from compositions of simulated lung fluid (Gamble's solution) and blood plasma.

| Phase | Gamble's Solution | | | Blood Plasma | | |
|---|---|---|---|---|---|---|
| | SI | log IAP | Log Ksp | SI | log IAP | log Ksp |
| Calcite | 1.12 | −7.44 | −8.56 | 0.70 | −7.86 | −8.56 |
| Dolomite | 2.10 | −15.26 | −17.36 | 1.43 | −15.93 | −17.36 |
| HAP | 7.77 | 3.32 | −4.45 | 6.52 | 2.08 | −4.45 |

Some 60% of the $CO_2$ in blood is transported as bicarbonate ($HCO_3^-$) [53], favoring crystallization of carbonates according to the reaction:

$$HCO_3^- + Ca^{2+} \leftrightarrow CaCO_3\downarrow + H^+ \qquad (3)$$

It has been shown experimentally that HAP solubility increases significantly with an increase in $P_{CO2}$, and that a small amount of calcite is precipitated at pH ca 7.4 and $P_{CO2}$ = 0.01–1.0 bar [54].

Apparently, the lower-lobe biochemical conditions are suitable for the precipitation and crystallization of both Mg-calcite and calcite in response to the dust deposition. Whether the mineralization of the upper lobes of lungs is dominated by calcium phosphates requires further investigation. We are not aware of a mineralogical study that would show the mineral composition of the apexes of lungs.

Age-related changes in lungs include, among others, progressive calcification of airways [55]. Based on the results of our study, it is tempting to hypothesize that age-related lung calcification is not only caused by natural physiological processes but is also driven by the accumulation of exogenous particles. It is possible that small (<3 cm in diameter) solid nodules commonly observed in the majority of elderly patients may have also developed on exogenous dust particles. Testing this hypothesis would require a cross-sectional examination of the nodules.

Exogenous particle-induced calcification is not the only cell-mediated mineralization in the alveolar region of lungs. Long (>10 μm) and thin (<0.5 μm) asbestos fibers cannot be entirely internalized by alveolar macrophages and, in a process called "frustrated phagocytosis," are coated with a thin layer of Fe-containing proteins, goethite and apatite [41,56]. However, our study showed that environmental exposure to mineral dust results principally in a large-scale particle-induced calcification of lungs. Precipitation of Mg-calcite and calcite on dust particles masks their occurrence in the lung. Thus, the actual number of inhaled dust particles deposited in lower lobes may be greater than observed because of the masking effect of carbonate coating.

## 6. Conclusions

While calcium salts precipitated in lungs are usually reported as phosphates (hydroxyapatite or whitlockite), calcium precipitates in RLL observed in this study were almost exclusively carbonates, with Mg-calcite predominant in the submicrometer size range, and calcite in larger grains and aggregates. Overall, Mg-calcite was the predominant carbonate in RLL, with an $MgCO_3$ content of <50 mol %. Thus, magnesium plays a significant role in lung mineralization, which has so far been overlooked. The medical term "calcification" is somewhat misleading, and does not reflect the actual chemical and mineralogical compositions of lung-endogenous mineralization. The calcium phosphate (hydroxyapatite) content in RLL was negligible. Lung fluids and blood plasma were saturated with respect to both apatite and calcite. Therefore, whether phosphate or carbonate will precipitate in the lungs depends on local conditions, namely $CO_2$ partial pressure. Elevated $P_{CO2}$ in lower lobes favors precipitation of carbonates.

Precipitation of Mg-calcite and calcite is, at least in part, induced by inhaled mineral dust particles that settled in the alveolar sacs. Insoluble dust particles serve as nucleation

sites for carbonate precipitates. The actual number of inhaled dust particles may be significantly larger than observed because of the masking effect of the carbonate coating. Our data support the observation of Churg and Wiggs [27] that total lower-lobe particle retention appears to be independent of the amount of smoking.

Three possible pathways for carbonates precipitation on dust particles are inferred from this study: (1) precipitation of ACC, followed by its transformation to calcite; (2) precipitation of Mg-ACC, followed by its transformation to Mg-calcite; and (3) precipitation of Mg-free ACC, followed by Mg-calcite heterogeneous nucleation and crystal growth.

**Author Contributions:** Conceptualization, J.J.; methodology, M.J. and J.J.; validation, M.J., J.J. and B.S.-K.; formal analysis, M.J. and J.J.; investigation, M.J. and B.S.-K.; resources, J.J.; data curation, M.J.; writing—original draft preparation, J.J.; writing—review and editing, J.J.; visualization, M.J.; supervision, J.J. All authors have read and agreed to the published version of the manuscript.

**Funding:** This work was financially supported by the statutory fund of the Institute of Earth Sciences, University of Silesia.

**Institutional Review Board Statement:** Ethical review and approval were waived for this study, because they were not required for autopsy samples provided by medical institutions.

**Informed Consent Statement:** Not applicable.

**Acknowledgments:** Valuable comments and suggestions by two anonymous reviewers that improved the final version of the manuscript are greatly appreciated. We are grateful to Padhraig Kennan for improving the English of the manuscript.

**Conflicts of Interest:** The authors declare no conflict of interest.

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
