# Peer review of "The Impact of Ambient Atmospheric Mineral-Dust Particles on the Calcification of Lungs"

_minerals, doi:10.3390/min11020125_

Round 1

Reviewer 1 Report

Dear Authors,

Congratulations on a fascinating manuscript!  I greatly enjoyed reading it and learned a lot from your observations. The manuscript is very good and well-prepared, and I have only minor comments. All comments (I think there are 92 comments) are marked in the edited pdf file attached to this message. 

This is an important contribution to medical mineralogy and should be published ASAP. 

Congratulations again and best wishes

Author Response

First of all, we thank the reviewer for his valuable comments, corrections, and suggestions. We implemented all of them and made corrections accordingly. We also greatly appreciate his very positive (not to say enthusiastic) remarks on our manuscript.

Answers to the reviewer’s specific comments.

line 152 – Table 1 summarizes previous study of the first author (MJ) by listing minerals constituting particles in the RLL tissues of 12 individuals without specifying their sex and age. Otherwise, the table would too large and served no purpose for the current article. For obtaining more detailed information on mineral composition of particles in female and male individuals (not much different), the interested reader if referred to the reference given in a Table 1. In the Materials section we specify the amount of female and male samples, because it is pertinent for further reading of the current article.

line 197 – the shape of the aggregates can be approximated by an oval; hence it made sense to measure their dimension as a diameter; whereas, single crystals are elongated and we measured their length along the longer axis.

line 239 - The mole percentage of MgCO3 in (Ca1-xMgx)CO3 was calculated knowing the molar Mg/Ca of 0.60 from Taunton et al. 2010. If, Mg/Ca = x/(1-x) = 0.60, then x = 0.375 and 1 – x = 0.625. From that 0.375/0.625 = 0.60.

line 245 – we inserted our short explanation for the steeper slope of the correlation line of females samples (in the text highlighted in yellow).

line 314 – these are our calculations (that is why we do not refer to any source of information in Table 3) done using the PHREEQC

Reviewer 2 Report

This well written study examined the mineral composition of lung lower lobe tissues of 12 individuals from Poland and observed that endogenous particles of Ca- and (Ca, Mg)-carbonates in lung tissues may originate from inhaled atmospheric dust particles.

The study is original since not that much is known about the mineral composition of non-occupational dust particles deposited in the lungs, but generalizability of results may be limited given the selection of only 12 individuals with a very specific exposition. Site characterization seems well performed, however less information is provided how the 12 USC residents were selected. Were these all autopsy samples available from the DiagnoMed company, or how was the sample size derived and the selection performed?

It is suggested that the amounts of carbonates might be lower in smoking individuals because smoking might decrease the pH of lung fluids ihibiting precipitation of carbonates, how does this influence emphysema formation in smokers? Why does elevated PCO2 in lower lobes of smokers does not favor precipitation of carbonates?

Does precipitation of magnesium-calcite in the lungs depends on dietary (or supplementary elevated) levels of magnesium?

The seasonality of peak ambient air pollution seemed reflected by the seasonal exacerbation of respiratory diseases, but what about exposition to pathogens? How does lung inflammation affect the mineral composition of lung lower lobe tissues?

What is the clinical impact of these results and what does it mean in terms of prevention or treatment?

Author Response

Reviewer: “The study is original since not that much is known about the mineral composition of non-occupational dust particles deposited in the lungs, but generalizability of results may be limited given the selection of only 12 individuals with a very specific exposition. Site characterization seems well performed, however less information is provided how the 12 USC residents were selected. Were these all autopsy samples available from the DiagnoMed company, or how was the sample size derived and the selection performed?”

We agree with reviewer that the generalization of results obtained during our study would be inappropriate and premature bearing in mind the limited number of samples from a single region. We avoided unjustified general statements. Nevertheless, we believe that our study contributes to the general knowledge on the mechanisms of lung response to the inhaled atmospheric dust particles. While mineral composition of mineral particles deposited in lungs is site-specific, the dust particles-induced precipitation of calcium and magnesium carbonates can be a common process.

Autopsy samples for our study were from the DiagnoMed company and from a clinical hospital, as stated in the article. Samples were selected based on the following criteria: (a) all individuals were born and lived through most of their life in the USC (individuals with unknown time of residence in the USC were excluded); (b) they did not suffer from lung diseases and they died from causes not related to lung disorder (natural causes or accidents); (c) they were not exposed occupationally to dust (with the exception of 2 coal miners, who however, showed no signs of dust-related lung disorder); (d) a suite of samples should represent individuals with a wide range of ages, translated into different dust accumulation time.

The sample size was a standard size of autopsy samples (as stated in the article) in accord with medical procedures in Poland.

The major factor limiting the amount of the material investigated was TEM. TEM observations of dust particles are labor-intensive and costly; hence only a limited number of samples could be observed during our study.

Reviewer: “It is suggested that the amounts of carbonates might be lower in smoking individuals because smoking might decrease the pH of lung fluids inhibiting precipitation of carbonates, how does this influence emphysema formation in smokers? Why does elevated PCO2 in lower lobes of smokers does not favor precipitation of carbonates?

We hypothesize that prolonged exposure to inhaled tobacco smoke (with pH around 6) decreases pH of lungs below the stability field of carbonates. In other words, carbonates cannot precipitate despite sufficiently high PCO2. Perhaps, other factors associated with the inhalation of tobacco smoke, including the influx of reactive oxygen species, e.g. H2O2 or (●OH), may also prevent precipitation of carbonates. Interestingly, hydroxyapatite is stable at mild acidic conditions (pH around 6), while carbonate apatite is not. The latter, likewise calcite has a stability field in an alkaline pH range. Apatite observed in our study show electron diffraction features typical of hydroxyapatite.

Emphysema in smokers is caused by chronic exposure to inhaled irritants that are present in tobacco smoke. There is a vast recent medical literature on mechanisms bringing about emphysema but this is beyond our expertise as mineralogists and geochemists.

Reviewer: “Does precipitation of magnesium-calcite in the lungs depends on dietary (or supplementary elevated) levels of magnesium?”

Magnesium is the second most abundant intracellural cation and is involved in more than 300 enzymatic reactions in the human body (e.g. Laires et al. Frontiers in Bioscience 9, 262-276, 2004). It is, therefore, not surprising to find Mg incorporated into lung mineralization. Kindly note, that (Ca,Mg)3PO4 and not apatite was reported by Belém et al. (2014) as chief constituent of metastatic calcification in human lungs. Perhaps, magnesium may have been overlooked in numerous analyses of calcified nodules and lesions, particularly if optical microscope was used for the phase identification. Alternatively, in some cases the amount of Mg may be below the detection limit of analytical instruments. We observed Mg-calcite only in submicrometer grains; while larger grains lacked the detectable magnesium. Perhaps, Mg was exhausted at the onset of carbonate precipitation.

There is no evidence for elevated dietary Mg levels in the USC. The bulk Mg content in the studied dry tissues is on average 422 ppm ranging from 371 and 377 ppm in individuals younger than 60 years to 468 ppm in older ones (Jabłońska, 2013). These values are not particularly high. While we do not know the dietary magnesium intake in the USC, there is no reason to believe that it significantly different from the rest of Europe. But this is an interesting question certainly worth pursuing the answer.

Reviewer: “The seasonality of peak ambient air pollution seemed reflected by the seasonal exacerbation of respiratory diseases, but what about exposition to pathogens? How does lung inflammation affect the mineral composition of lung lower lobe tissues?”

There have been only few papers on the exposition to indoor air pathogens in the USC (e.g. Pastuszka et al. 2000, Atmos. Environ. 34, 3833-3842). We are not aware of publications related to the role of airborne pathogens in respiratory diseases in the USC. Most researchers are preoccupied with the impact of inorganic aerosols on human health. We recently acquired an equipment to collect bioaerosol aiming at studying its role in air pollution.

This is an important question that require serious investigation beyond the scope of our present study. The reactive oxygen species generated during the inflammatory response to oxidative stress caused by the inhaled dust particles may affect redox-sensitive minerals. This has been observed in the so-called frustrated phagocytosis of long asbestos fibers, which were coated by Fe(III)-proteins and FeOOH. To our knowledge nobody has actually looked at this problem.

Reviewer: “What is the clinical impact of these results and what does it mean in terms of prevention or treatment?”

Perhaps, we as mineralogists and geochemists, are not in a position to answer this question. The fact is, that there is an interest in calcification of lungs from the medical point of view. However, despite that interest, there have been only a few attempts to actually identifying the mineral composition of calcium lesions and nodules. In our paper, we show a new role of the inhaled dust particles in inducing lower lobe calcification in the form of carbonates. We also speculate that the age-related progressive calcification of lungs is not only a normal physiological process but, it may be enhanced by precipitation of calcium(magnesium) carbonates induced by progressively accumulated inhaled dust particles. Whether our findings are relevant for clinicians or not, we leave to their judgement.